# Genomic profiling in hidradenitis suppurativa: InterOmics pipeline for DNA-RNA sequencing highlights HLA variants, keratin-associated mutations and extracellular matrix alterations as contributing factors to HS pathogenesis

Lucas André Cavalcanti Brandão[1,2], Ronald Rodrigues de Moura[3], Bruno Rodrigo Assunção[1,2], Cecilia Del Vecchio[3], Adamo Pio d'Adamo[3,4], Gudrun Ratzinger[5], Barbara Böckle[5], Nina Frischhut[5], Wolfram Jaschke[5], Matthias Schmuth[5], Muhammad Suleman[6], Angelo Valerio Marzano[7,8], Chiara Moltrasio[7], Paola Maura Tricarico[9], Sergio Crovella[10]*

1 Molecular Pathology Laboratory, Pathology Department, Federal University of Pernambuco, Recife, Brazil, 2 OneHealth Laboratory, Keizo Asami Institute (iLIKA), Recife, Brazil, 3 Department of Medical Surgical and Health Sciences, University of Trieste, Trieste, Italy, 4 Department of Advanced Diagnostics, Institute for Maternal and Child Health, IRCCS "Burlo Garofolo", Trieste, Italy, 5 Department of Dermatology, Venereology and Allergy, Medical University of Innsbruck, Innsbruck, Austria, 6 Laboratory of Animal Research Center (LARC), Qatar University, Doha, Qatar, 7 Dermatology Unit, Fondazione IRCCS Ca' Granda Ospedale Maggiore Policlinico, Milan, Italy, 8 Department of Pathophysiology and Transplantation, Università degli Studi di Milano, Milan, Italy, 9 Department of Pediatrics, Institute for Maternal and Child Health - IRCCS Burlo Garofolo, Trieste, Italy, 10 Department of Biomedical Sciences, Qatar University, Doha, Qatar

* sgrovella@qu.edu.qa

## Abstract

Hidradenitis suppurativa (HS) is a chronic autoinflammatory skin disorder with a complex genetic and molecular basis. To advance its characterization, we applied InterOmics, a novel bioinformatics pipeline integrating whole exome sequencing (WES) and RNA sequencing (RNA-seq), to saliva and skin biopsy samples from six HS patients. This approach enabled a comprehensive multiomics investigation, identifying disease-associated genetic variants and transcriptomic alterations. A key innovation of InterOmics is the Multiomics Variant Category, which classifies variants based on DNA and RNA data, capturing regulatory mechanisms such as allele-specific expression, RNA editing, nonsense-mediated decay, and gain-of-function mutations. Our findings highlight HLA gene variants and keratin-related mutations as potential contributors to HS pathogenesis. By bridging genomic and transcriptomic data, InterOmics enhances variant interpretation. This study underscores the power of multiomics-driven approaches in deciphering complex diseases, paving the way for precision medicine in HS.

**Data availability statement:** All relevant data are within the manuscript and its Supporting Information files. However, the whole exome sequencing (WES) and RNA sequencing (RNA-seq) data generated in this study are available upon reasonable request. Due to ethical restrictions concerning human genomic data privacy, raw sequencing files cannot be publicly shared but can be accessed through a controlled request process. Researchers interested in obtaining the data should contact the corresponding author (sgrovella@qu.edu.qa) and provide documentation of institutional ethical approval for human genetic research. Data access will be granted upon agreement to comply with ethical and privacy regulations governing human genomic data sharing.

**Funding:** Our study has been supported by the project EraPerMed 2018-137 BATMAN (Biomolecular Analyses for Tailored Medicine in Acne iNversa) project, by the project CNPq/MCTI Nº 10/2023 – UNIVERSAL from the Brazilian Council for Research and by the project RC16/18 (Ricerca Corrente) from IRCCS Burlo Garofolo, Trieste, Italy. The funders had no role in study design, data collection and analysis, decision to publish, or preparation of the manuscript."

## Introduction

Hidradenitis suppurativa (HS) is a chronic, autoinflammatory skin disorder characterized by painful nodules, abscesses, and interconnected tunnels in apocrine gland-bearing areas. Affecting approximately 1–4% of the global population, HS has a complex pathogenesis involving both genetic predisposition and environmental triggers such as smoking, obesity, and mechanical friction [1]. Familial clustering, with about 35% of patients reporting a family history, suggests a strong hereditary component in disease susceptibility [2]. Despite advances in clinical management, the heterogeneous nature of HS and variable treatment responses highlight the need for a deeper molecular understanding to improve patient stratification and therapeutic strategies.

The rapid evolution of next-generation sequencing (NGS) technologies has significantly advanced our ability to profile the molecular mechanisms underlying complex diseases. Whole exome sequencing (WES) and whole genome sequencing (WGS) are now widely used to identify genetic abnormalities, including single nucleotide variants (SNVs) and copy number variations (CNVs), that contribute to disease pathogenesis [3]. On the other hand, RNA sequencing (RNA-seq) provides a dynamic snapshot of the transcriptome, offering insights into gene expression patterns, alternative splicing events, and post-transcriptional modifications that are crucial for understanding disease biology [4–7].

In the context of HS, transcriptomic analyses have largely focused on differentially expressed genes (DEGs) to identify dysregulated pathways involved in inflammation, tissue remodeling, and immune responses [8–11]. While DEG analysis remains valuable, recent advances in NGS technology have expanded RNA-seq applications beyond gene expression profiling, enabling the detection of genetic variants, transcript isoforms, and RNA editing events [12]. However, variant detection in RNA-seq data presents unique challenges. The variability in transcript isoforms, influenced by alternative splicing, makes it difficult to distinguish true genetic variants from isoform-specific alterations [13]. RNA editing, particularly adenosine-to-inosine (A-to-I) modifications, can also be misinterpreted as SNVs, complicating the analysis and interpretation of sequencing data [14]. Additionally, genes with low expression levels may have insufficient sequencing coverage, increasing background noise and making it more challenging to confidently call SNVs [15].

Given these challenges, integrating DNA and RNA sequencing data provides a more comprehensive and accurate molecular characterization of disease-associated variants. Here, we introduce InterOmics, a bioinformatics pipeline specifically designed to combine WES and RNA-seq data for a multiomic analysis of genetic and transcriptomic variations. InterOmics implements a systematic variant classification approach, linking genetic mutations with gene expression changes to uncover regulatory mechanisms that may drive disease phenotypes.

In this study, we applied InterOmics to DNA and RNA sequencing data from six HS patients, aiming to establish a structured framework for investigating the genetic basis of HS and its impact on transcriptomic profiles. By leveraging multiomic

integration, we seek to identify novel molecular signatures associated with HS and explore potential pathways for personalized therapeutic interventions. This approach represents a significant step toward precision medicine applications in HS, providing a scalable model for studying other complex inflammatory diseases.

## Materials and methods

### Study design, patients and sequencing

This study was approved by the Ethical Committee "Comitato Etico Unico Regionale (CEUR) of Friuli Venezia Giulia (FVG) (RC 16/18, Prot. N.0001094 (14/01/2019), CEUR-2018-Sper-127-BURLO IRCCS Burlo Garofolo" Italy. This Ethical Committee also covered the enrollment of patients from the European Consortium (Italy, France, Austria, France and Slovenia) for tailored diagnosis of patients affected by Hidradenitis suppurativa, EraPerMed2018–137. The study included six unrelated Austrian patients (4 men and 2 women, median age 41.5 years, range 29–67 years) diagnosed with moderate-to-severe HS. All participants provided written informed consent prior to sample collection..All participants underwent saliva sampling and surgical removal of HS lesional areas through an 8 mm punch biopsy (Kai Medical, Seki City, Gifu, Japan), in January 2024, at the Department of Dermatology, Venereology, and Allergology, Medical University of Innsbruck, Austria.

The mRNA was extracted from lesional skin biopsies of the patient enrolled, utilizing the RNeasy Lipid Tissue Mini Kit (Qiagen, Milano, Italy) as per the manufacturer's instructions. This was followed by the extraction of genomic DNA from saliva samples with the Oragene-DNA (Oragene®, Ottawa, Canada) kit, also adhering to the manufacturer's guidelines. Both DNA and mRNA samples were then sent to Macrogen Europe (Amsterdam, Netherlands) for sequencing.

The Exome Sequencing Analysis achieved an average coverage of 150x, employing the Illumina® SureSelect Human V7 Kit (San Diego, CA, USA) for library preparation and sequencing on the Illumina® HiSeq 2500 System (San Diego, CA, USA). This process produced paired end reads of 150 base pairs in length. For RNA-seq, we sequenced an average of 60 million paired-end reads using the Illumina® TruSeq Stranded Library. The quality of both sequencing outputs was assessed using FastQC on the raw sequencing files (fastq.gz format), evaluating parameters such as average read length, quality scores for reads and bases, and the presence of adapters.

### Whole Exome Sequencing (WES)

The WES reads were analyzed following the good practices of Broad Institute [16]. In summary, the residual adapters, short reads (below 25 base pairs) and/or low-quality reads (Q<20) were removed using TrimGalore (Version 0.6.5 https://www.bioinformatics.babraham.ac.uk/projects/trim_galore/). Unmapped reads were aligned with NCBI Genome Reference Consortium Human Build 38 (GRCh38) as human reference genome using Burrows-Wheeler Aligner (BWA) algorithm [17]; marking and removal of duplicated reads as well as base quality score recalibration was carried out using Picard Tools v.2.7.0 (https://broadinstitute.github.io/picard/) and GATK v.4.1.2.0 [18].

### RNA sequencing (RNA-Seq)

The reads from the raw sequencing files were processed using Trimmomatic software v.0.39 [19] to trim Illumina adapters and to exclude reads counting fewer than 25 bases. Then, the remaining reads were mapped on the National Center for Biotechnology (NCBI) human GRCh38 reference genome and sorted by coordinates using STAR aligner v. 2.7.6a [20]. Then, the mapped reads were counted using the gene quantification function, which can link the mapped read to its respective gene. The count is concluded by its normalization. Both methods were calculated using DESeq2 v. 1.40.2 [21]. Gene and isoform transcript expressions were calculated for each individual using RSEM v.1.3.3 [22].

### SNV and small insertion and deletion (Indel) detection

Variant calling was made using Strelka2.9.10 [23], where both SNVs and indels were filtered out using the following parameters: QD < 2.0, FS > 60.0, MQ < 40.0, MQ Rank Sum < −12.5, Read Pos Rank Sum < −8.0 and GQ < 20.0, for SNPs; and QD < 2.0, FS > 200.0, Read Pos Rank Sum < −20.0 and GQ < 20.0, for indels. Variant annotation was made using Annovar software v.2019Oct24 [24].

Furthermore, we did a 'in silico reverse transcription' (from now on, cDNA dataset) of the mapped RNA-seq bam files using the same steps we did for WES analysis as well as variant calling and annotation, with the addition of one step to splitting N-CIGAR reads into multiple alignments after marking and removing duplicated reads.

### Integrating WES and RNA-Seq data

We used GRCh38 as a reference genome for both whole exome sequencing (WES) and RNA sequencing (RNA-seq) alignment. Our approach was designed to balance consistency, transcriptome completeness and accurate variant detection. This choice ensures consistency in variant calling and transcript quantification, providing a standardized reference point for comparing genetic and transcriptomic variations across patients. The pipeline generates a WES-derived variant profile by mapping sequencing reads to GRCh38, identifying deviations from this reference genome in each individual.

In parallel, RNA-seq reads are also aligned to GRCh38-based transcript annotations, allowing us to quantify gene expression and identify transcriptomic variants such as allele-specific expression (ASE), i.e., variants that is heterozygous in the WES data, but homozygous for the alternative allele in the RNA-seq data, and RNA editing (RNAe) events, which consists of variants that are homozygous for the alternate allele in the WES data and heterozygous in the RNA-seq data.

Also, with both WES and RNA-seq data it is possible to infer the occurrence of Nonsense-mediated mRNA decay (NMD) [25], based on the expression of the genes. Substantially, we could expect a gene being targeted as NMD if there is a premature stop codon or a frameshift mutation in a gene found to be less or not expressed in the patient in comparison to a reference expression for the same tissue. In this case, we obtained the transcript per million (TPM) values for skin not exposed to the sun from GTex database using R "hpar" package v.1.48.0 [26].

Similarly, one can also infer Gain-of-Function (GoF) mutations by selecting non-intronic, non- intergenic, non-synonymous SNV nor frameshift INDEL variants that has medium or high expression in the patient, while it is below cutoff or lower expressed in the reference expression dataset.

These four categories are classified as Multiomics Variants, since their assignment depends on both WES and RNA-seq data.

## Results and discussion

InterOmics workflow integrates genomic and transcriptomic data through a systematic three-stage process. In the first stage, raw sequencing files undergo quality control, cleaning, alignment, and annotation, establishing a foundation for reliable variant detection. The second stage organizes this processed data in PosgreSQL database (v.16.0), enabling efficient data retrieval and analysis. The third stage implements the integration protocol, designed to obtain an individual multiomics variation profile (Fig 1).

The integration of WES and RNA sequencing data yielded high-quality genomic information, with approximately 98% of reads successfully aligning to the reference genome. Table 1 shows the distribution of WES and RNA-seq variants in the HS patients.

ASE and RNAe variants, though less numerous, could provide insights into gene regulation in HS [27]. We identified some ASE and RNAe variants in genes involved in inflammatory responses, especially in the HLA *loci*. S1 and S2 Tables in S1 File show all ASE and RNAe variants, respectively, where Tables 2 and 3 show selected ASE and RNAe variants, respectively, after applying the following filters:

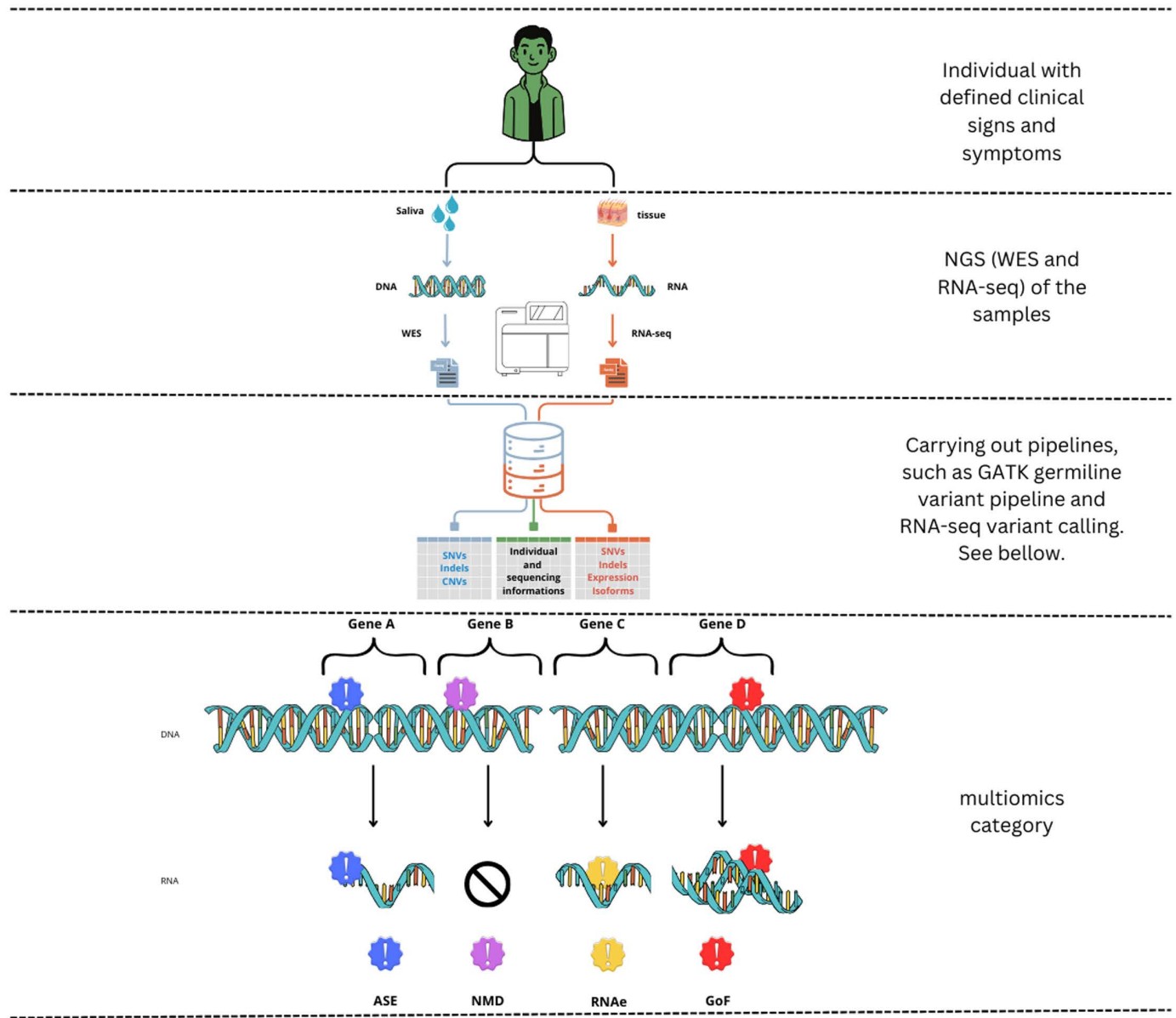

**Fig 1. Interomics pipeline: A step-by-step approach to personalized genomic profiling.** The InterOmics pipeline begins with the collection of biological samples from the individual being tested. It then progresses to the simultaneous analysis of Whole Exome Sequencing (WES) and RNA Sequencing (RNASeq). Following this parallel analysis, the data are integrated to identify Single Nucleotide Variants (SNVs), Copy Number Variants (CNVs), insertions and deletions (indels), and expression isoforms. This comprehensive approach culminates in the creation of a personalized genomic and transcriptomic profile for the individual, offering detailed insights into their unique genetic makeup. In the Figure, The Multiomics Category variants are: Alelle specific expression (ASE); Predicted Nonsense-mediated decay (NMD); RNA editing events (RNAe); and Predicted gain-of-function (GoF).

1. excluding intronic, intergenic, upstream and downstream variants;

2. excluding synonymous SNV;

3. excluding variants with minor allele frequency (MAF) > 1%.

**Table 1. Variants count according to NGS (WES and RNA-seq) findings in the six HS patients.** The counts for the multiomics categories (ASE and RNAe) are also shown. WES: whole exome sequencing; RNA-seq: RNA sequencing; ASE: Allele Specific Expression; RNAe: RNA editing events; NDM: Nonsense-mediated mRNA decay; GoF: Gain-of-function variant.

| Category | AUT101 | AUT104 | AUT108 | AUT201 | AUT401 | AUT402 |
|---|---|---|---|---|---|---|
| **WES** | 99,109 | 104,670 | 127,709 | 101,958 | 100,015 | 95,594 |
| **RNA-seq** | 31,149 | 30,156 | 74,135 | 15,758 | 26,855 | 20,031 |
| ASE | 8 | 28 | 26 | 20 | 26 | 9 |
| RNAe | 40 | 42 | 54 | 32 | 29 | 26 |
| NMD | 14 | 9 | 13 | 6 | 17 | 10 |
| GoF | 11 | 14 | 14 | 19 | 15 | 14 |

**Table 2. List of filtered Allele specific expression (ASE) variants, i.e., only the alternative (alt) allele was found to be expressed, among the patients.**

| Sample | chr | Start base | End base | Ref | Alt | Gene name | Gene location | Variant consequence | MAF | rs id | TPM (gtex) | TPM (sample) |
|---|---|---|---|---|---|---|---|---|---|---|---|---|
| AUT101 | chr6 | 29888222 | 29888222 | G | C | HLA-H | exonic ncRNA | – | – | rs78277628 | 4019.991 | 8706.852 |
| AUT104 | chr6 | 32584352 | 32584354 | CCC | – | HLA-DRB1 | exonic | nonframeshift deletion | 0.00480 | rs745615811 | 8603.733 | 6453.473 |
| AUT104 | chr6 | 32584355 | 32584355 | – | ATA | HLA-DRB1 | exonic | nonframeshift insertion | 0.00500 | rs758070868 | 8603.733 | 6453.473 |
| AUT108 | chr6 | 32589775 | 32589775 | A | C | HLA-DRB1 | UTR5 | – | 0.00380 | rs1059546 | 8603.733 | 9952.231 |
| AUT108 | chr6 | 32589702 | 32589702 | G | A | HLA-DRB1 | exonic | nonsynonymous SNV | 0.00090 | rs9270302 | 8603.733 | 9952.231 |
| AUT201 | chr6 | 32579025 | 32579025 | C | A | HLA-DRB1 | UTR3 | – | – | rs3180799 | 8603.733 | 14435.322 |
| AUT201 | chr6 | 32589702 | 32589702 | G | A | HLA-DRB1 | exonic | nonsynonymous SNV | 0.00090 | rs9270302 | 8603.733 | 14435.322 |
| AUT401 | chr6 | 29888222 | 29888222 | G | C | HLA-H | exonic ncRNA | – | – | rs78277628 | 4019.991 | 4054.98 |
| AUT402 | chr6 | 32579075 | 32579076 | TG | – | HLA-DRB1 | UTR3 | – | 0.00003 | rs760049502 | 8603.733 | 3296.966 |
| AUT402 | chr6 | 32589702 | 32589702 | G | A | HLA-DRB1 | exonic | nonsynonymous SNV | 0.00090 | rs9270302 | 8603.733 | 3296.966 |
| AUT402 | chr3 | 49358244 | 49358246 | GCC | – | GPX1 | exonic | nonframeshift deletion | – | – | 2629.324 | 1459.556 |

A third insight obtained by integrating WES and RNA-seq was the inference of NMD. In Table 4, we summarize the variants possibly causing NMD. The list with all NMD variants can be seen in S3 Table in S1 File, while Table 4 shows only variants with MAF < 0.01. All five variants with MAF < 0.01 were heterozygous, with two of them being in genes directly associated with keratin formation (rs2857667 and rs200049107), and the others are present in genes associated with cell metabolism.

A fourth category are the possible GoF variants. Table 5 shows the number of GoF variants per sample, while the S4 Table in S1 File displays the 87 GoF found among the patients with MAF < 0.01. Patient AUT108 has one heterozygous GoF deletion in the 3' UTR of the *COL19A1* gene (rs36112821), therefore having a higher expression for this gene in the skin.

**Table 3. List of filtered RNA editing (RNAe) variants, i.e., the genotype is heterozygous in the RNA but homozygous for the alternative (alt) allele in the DNA, was found to be expressed, among the patients.**

| Sample | chr | Start base | End base | Ref | Alt | Gene name | Gene location | Variant consequence | MAF | rs id | TPM (gtex) | TPM (sample) |
|---|---|---|---|---|---|---|---|---|---|---|---|---|
| AUT101 | chr6 | 31356751 | 31356751 | G | A | HLA-B | exonic | stopgain | – | rs1071817 | 39269.17 | 50423.15 |
| AUT101 | chr6 | 31356749 | 31356749 | C | G | HLA-B | exonic | nonsynonymous SNV | 0.0033 | rs1131212 | 39269.17 | 50423.15 |
| AUT101 | chr6 | 31356750 | 31356750 | T | G | HLA-B | exonic | nonsynonymous SNV | – | rs1140404 | 39269.17 | 50423.15 |
| AUT101 | chr6 | 31356717 | 31356717 | A | G | HLA-B | exonic | nonsynonymous SNV | 0.0000 | rs9266161 | 39269.17 | 50423.15 |
| AUT101 | chr6 | 31356718 | 31356718 | G | C | HLA-B | exonic | nonsynonymous SNV | 0.0000 | rs9266162 | 39269.17 | 50423.15 |
| AUT104 | chr6 | 32517690 | 32517690 | A | T | HLA-DRB5 | UTR3 | – | 0.0003 | rs113473719 | 3398.56 | 2722.61 |
| AUT104 | chr6 | 31271165 | 31271165 | G | A | HLA-C | exonic | nonsynonymous SNV | 0.0006 | rs2308590 | 24170.05 | 8623.30 |
| AUT104 | chr6 | 31271165 | 31271165 | G | T | HLA-C | exonic | nonsynonymous SNV | 0.0068 | rs2308590 | 24170.05 | 8623.30 |
| AUT104 | chr6 | 31271153 | 31271153 | A | C | HLA-C | exonic | nonsynonymous SNV | 0.0037 | rs2308592 | 24170.05 | 8623.30 |
| AUT104 | chr6 | 31356717 | 31356717 | A | G | HLA-B | exonic | nonsynonymous SNV | 0.0000 | rs9266161 | 39269.17 | 20791.74 |
| AUT104 | chr6 | 31356718 | 31356718 | G | C | HLA-B | exonic | nonsynonymous SNV | 0.0000 | rs9266162 | 39269.17 | 20791.74 |
| AUT108 | chr6 | 31271165 | 31271165 | G | T | HLA-C | exonic | nonsynonymous SNV | 0.0068 | rs2308590 | 24170.05 | 11304.97 |
| AUT108 | chr6 | 31271165 | 31271165 | G | A | HLA-C | exonic | nonsynonymous SNV | 0.0006 | rs2308590 | 24170.05 | 11304.97 |
| AUT108 | chr6 | 31271153 | 31271153 | A | C | HLA-C | exonic | nonsynonymous SNV | 0.0037 | rs2308592 | 24170.05 | 11304.97 |
| AUT401 | chr6 | 32641522 | 32641522 | A | C | HLA-DQA1 | exonic | nonsynonymous SNV | 0.0020 | rs1064944 | 3647.17 | 3994.08 |
| AUT401 | chr6 | 32641522 | 32641522 | A | G | HLA-DQA1 | exonic | nonsynonymous SNV | 0.0000 | rs1064944 | 3647.17 | 3994.08 |
| AUT401 | chr6 | 31270482 | 31270482 | G | T | HLA-C | exonic | nonsynonymous SNV | 0.0012 | rs1131096 | 24170.05 | 12346.23 |
| AUT402 | chr6 | 31270482 | 31270482 | G | T | HLA-C | exonic | nonsynonymous SNV | 0.0012 | rs1131096 | 24170.05 | 5888.60 |

**Table 4. List of NMD variants among the patients with minor allele frequency (MAF) lower than 1%.**

| Sample | chr | Start base | End base | Ref | Alt | Gene name | Variant consequence | MAF | rs id | TPM (gtex) | TPM (sample) |
|---|---|---|---|---|---|---|---|---|---|---|---|
| AUT101 | chr12 | 52317765 | 52317765 | G | T | KRT83 | stopgain | 0.0069 | rs2857667 | 241.31 | 0.00 |
| AUT101 | chr16 | 31428324 | 31428324 | A | G | COX6A2 | startloss | 0.0005 | rs200780049 | 118.48 | 0.00 |
| AUT101 | chr15 | 75293804 | 75293804 | C | T | GOLGA6D | stopgain | 0.0013 | rs201679690 | 141.83 | 7.34 |
| AUT101 | chr16 | 67884906 | 67884906 | G | A | NRN1L | startloss | 0.0002 | rs201174409 | 14.67 | 4.40 |
| AUT104 | chr17 | 41065564 | 41065564 | C | T | KRTAP2–4 | stopgain | 0.0031 | rs200049107 | 106.60 | 5.90 |

**Table 5. Number of possible Gain-of-Function (GoF) variants among the samples, according to their location in the gene.**

| Sample | Upstream | UTR5 | Exonic | Exonic ncRNA | Splicing | UTR3 | Downstream |
|--------|----------|------|--------|--------------|----------|------|------------|
| AUT101 | 1 | 2 | 2 | 3 | – | 3 | – |
| AUT104 | 4 | 1 | 1 | 7 | – | – | 1 |
| AUT108 | 2 | 2 | 1 | 1 | – | 8 | – |
| AUT201 | 4 | 3 | 2 | 5 | – | 4 | 1 |
| AUT401 | – | 1 | 2 | 5 | 1 | 4 | 2 |
| AUT402 | 2 | 1 | 1 | 8 | – | 2 | – |

Numerous frameworks have been developed for the joint analysis of DNA and RNA sequencing data, each tailored to specific study designs and objectives. Expression Quantitative Trait Loci (eQTL) analysis pipelines (e.g., Matrix eQTL, FastQTL) seek to associate genetic variants with transcript abundance but typically require very large sample sizes (n > 200) to achieve sufficient statistical power and to adjust for multiple testing and population structure confounders. Similarly, unsupervised multi-omics clustering frameworks like MOFA and iClusterPlus are highly effective for large heterogeneous cohorts, enabling the discovery of disease subtypes or novel biological pathways by extracting latent factors across omics layers.

However, these tools are not easily applicable to small-scale, clinical investigations due to their sample size requirements and underlying assumptions. InterOmics was explicitly designed to address this gap, providing a variant-centric framework that enables individual-level genomic-transcriptomic integration. Rather than performing association analyses or unsupervised clustering, InterOmics classifies genetic variants based on their transcriptional consequences—such as allele-specific expression, RNA editing, nonsense-mediated decay, or gain-of-function—within the same subject.

This focus allows InterOmics to be particularly useful for small sample-size studies typical of pilot projects, rare disease cohorts, or early translational research. It tolerates sample sizes that would render statistical association methods underpowered and instead emphasizes biologically interpretable integration designed to inform hypothesis generation and clinical decision-making. Therefore, InterOmics complements existing frameworks, offering a unique solution when the research context involves a limited number of deeply characterized individuals.

The integration of whole exome sequencing (WES) and RNA sequencing (RNA-seq) has advanced our ability to analyze both genetic variation and gene expression, yet the clinical application of multiomics strategies remains limited. In this study, we introduced InterOmics, a systematic pipeline designed to merge WES and RNA-seq data, providing a more comprehensive characterization of disease-associated variants in HS. Unlike widely used multiomic approaches such as iCluster and MOFA, which focus on clustering molecular profiles, InterOmics was specifically developed to integrate genomic and transcriptomic data within a biological context, enabling a more refined classification of variants and their regulatory impact.

A key innovation of this study is the Multiomics Variant Category system, which categorizes variants based on their detection in both DNA and RNA data, allowing the identification of functionally relevant alterations. Our results reveal that allele-specific expression (ASE) variants were predominantly found in HLA genes, supporting recent evidence that specific HLA alleles may modulate susceptibility to HS [28,29]. Additionally, RNA editing (RNAe) variants, particularly adenosine-to-inosine (A-to-I) conversions, were identified, further emphasizing post-transcriptional regulation in HS pathogenesis.

Furthermore, we identified nonsense-mediated mRNA decay (NMD) variants associated with keratin-related genes (*KRT83* and *KRTAP2–4*). These variants introduce premature stop codons, likely impairing keratin protein synthesis, which may contribute to tissue remodeling and skin barrier dysfunction in HS. Furthermore, these findings support the proposed concept of HS as an autoinflammatory keratinization disease (AiKD) [30].

Conversely, gain-of-function (GoF) variants, particularly in regulatory regions such as the 3' UTR, were observed. Notably, a single base deletion (rs36112821) in *COL19A1*, a key collagen-related gene, was linked to altered transcript stability, reinforcing the role of extracellular matrix remodeling in HS [31].

Our findings highlight the importance of integrating WES and RNA-seq data to capture not only genetic mutations but also their transcriptional consequences. The high concordance between genetic and transcriptomic alterations observed in this study suggests that certain variants actively modulate gene expression, emphasizing their potential functional relevance in HS pathogenesis.

Despite these promising insights, some limitations must be acknowledged. The relatively small patient cohort restricts the generalizability of our findings, highlighting the need for larger validation studies. Additionally, while InterOmics provides a robust framework for variant classification, functional assays will be essential to confirm the biological impact of the identified variants and their relevance to HS pathogenesis.

## Conclusions

This study demonstrates the power of multiomics integration in deciphering the genetic and transcriptomic landscape of HS. The InterOmics pipeline enabled a systematic classification of genomic and transcriptomic variants, revealing novel molecular signatures associated with HS pathogenesis. Our findings underscore the role of HLA variants, keratin-associated mutations, and extracellular matrix alterations in disease progression, reinforcing the utility of multiomic-driven approaches in complex inflammatory disorders.

While this study establishes a foundation for integrating DNA and RNA sequencing in HS research, further investigations are necessary to translate these findings into clinical applications. Expanding the patient cohort will be critical to distinguishing disease-driving variants from individual-specific variations. Additionally, functional validation studies are essential to confirm the biological significance of multiomic variants and their potential as biomarkers or therapeutic targets.

All in all, our findings highlight the transformative potential of multiomics approaches in understanding complex diseases like HS.

## Supporting information

**S1 File. S1 Table. Allele specifi expressed variants.** List of Allele Specific Expressed (ASE) variants with their location at both chromosomal and gene level, reference and alternat alleles, variant consequence, minor allele frequency (MAF), dbSNP (rs) id, base quality and depth metrics, Transcript per million (TPM) values for both GTex skin (not exposed to sun) gene expression and sample. **S2 Table. RNA editing**. List of RNA editing (RNAe) variants with their location at both chromosomal and gene level, reference and alternat alleles, variant consequence, minor allele frequency (MAF), dbSNP (rs) id, base quality and depth metrics, Transcript per million (TPM) values for both GTex skin (not exposed to sun) gene expression and sample. **S3 Table. Nonsense-mediated messenger RNA**. List of inferred Nonsense-mediated messenger RNA (mRNA) decay (NMD) variants with their location at both chromosomal and gene level, reference and alternat alleles, variant consequence, minor allele frequency (MAF), dbSNP (rs) id, base quality and depth metrics, Transcript per million (TPM) values for both GTex skin (not exposed to sun) gene expression and sample. The expression categories are also showed. **S4 Table: Gain of function variants**. List of inferred Gain of Function (GoF) variants with their location at both chromosomal and gene level, reference and alternat alleles, variant consequence, minor allele frequency (MAF), dbSNP (rs) id, base quality and depth metrics, Transcript per million (TPM) values for both GTex skin (not exposed to sun) gene expression and sample. The expression categories are also shown.
(ZIP)

## Acknowledgments

Fig 1 has been designed using resources from canva.com. Reference manager was accomplished by Mendeley Reference Manager.

## Author contributions

**Conceptualization:** Lucas André Cavalcanti Brandão, Paola Maura Tricarico, Sergio Crovella.

**Data curation:** Lucas André Cavalcanti Brandão, Ronald Rodrigues de Moura, Wolfram Jaschke.

**Formal analysis:** Adamo Pio d'Adamo.

**Investigation:** Cecilia Del Vecchio, Chiara Moltrasio, Paola Maura Tricarico, Sergio Crovella.

**Methodology:** Lucas André Cavalcanti Brandão, Ronald Rodrigues de Moura.

**Resources:** Gudrun Ratzinger, Barbara Böckle, Nina Frischhut, Matthias Schmuth.

**Software:** Ronald Rodrigues de Moura, Bruno Rodrigo Assunção, Muhammad Suleman.

**Supervision:** Matthias Schmuth, Angelo Valerio Marzano.

**Writing – original draft:** Paola Maura Tricarico, Sergio Crovella.

**Writing – review & editing:** Lucas André Cavalcanti Brandão, Cecilia Del Vecchio, Chiara Moltrasio, Sergio Crovella.

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
