## [Decision Letter · Decision Letter 0]

Dear Dr. Crovella,

Thank you for submitting your manuscript to PLOS ONE. After careful consideration, we feel that it has merit but does not fully meet PLOS ONE’s publication criteria as it currently stands. Therefore, we invite you to submit a revised version of the manuscript that addresses the points raised during the review process.

We look forward to receiving your revised manuscript.

Kind regards,

Donatella Mentino

Academic Editor

PLOS ONE

Journal Requirements:

“Our study has been supported by the project EraPerMed 2018-137 BATMAN (Biomolecular Analyses for Tailored Medicine in Acne iNversa) project, by the project CNPq/MCTI Nº 10/2023 – UNIVERSAL from the Brazilian Council for Research and by the project RC16/18 (Ricerca Corrente) from IRCCS Burlo Garofolo, Trieste, Italy.”

4. Please note that funding information should not appear in the Acknowledgments section or other areas of your manuscript. We will only publish funding information present in the Funding Statement section of the online submission form. Please remove any funding-related text from the manuscript. 

7.  We note you have included a table to which you do not refer in the text of your manuscript. Please ensure that you refer to Tables 2 to 4 in your text; if accepted, production will need this reference to link the reader to the Table.

8. We note that there is identifying data in the Supporting Information file < TableS2_PONE.xls>. Due to the inclusion of these potentially identifying data, we have removed this file from your file inventory. Prior to sharing human research participant data, authors should consult with an ethics committee to ensure data are shared in accordance with participant consent and all applicable local laws.

-Location data

Please remove or anonymize all personal information (SNP_Id, providerId) ensure that the data shared are in accordance with participant consent, and re-upload a fully anonymized data set. Please note that spreadsheet columns with personal information must be removed and not hidden as all hidden columns will appear in the published file.

**Additional Editor Comments:**

I would like to thank you for your work and for the significant contribution it makes in the field of genomics and the integration of multiomic data. However, I have some observations and questions that could help contribute to a greater clarity and understanding of your study. Additionally, I believe that Major revisions are necessary, and it is important to expand the manuscript by addressing the reviewers' comments in detail.

Clarification of the Objective: Clearly define whether the main objective is the introduction of the InterOmics tool or the analysis of Hartnup syndrome (HS).Details on InterOmics: Provide information on where to access and install InterOmics, such as a GitHub repository.Explanation of the Algorithm Steps: Offer a detailed description of the logic and objectives of each phase of the protocol in the methods section.Clarity on TPM Cutoff: Explain the choice of the 10 TPM expression cutoff and its relevance to the results.Comparison with Other Tools: Discuss the advantages and differences of InterOmics compared to other existing technologies in the multiomic landscape.Risks of False Positives: Elaborate on how the risks of false positives are managed, especially regarding the increase in identified variants.Connection to HS Pathogenesis: Provide more detailed explanations on how the results contribute to a better understanding of the mechanisms of HS.Data Accessibility: Offer key guidance on how readers can access patient data and the requirements for consultation.Correction of References and Proofreading: Review inconsistencies in the text and correct any language and writing errors.These points can help the authors improve the clarity and quality of their manuscript.

Reviewers' comments:

Reviewer's Responses to Questions

**Comments to the Author**

1. Is the manuscript technically sound, and do the data support the conclusions?

Reviewer #1: Partly

Reviewer #2: Partly

Reviewer #3: Yes

Reviewer #4: Yes

2. Has the statistical analysis been performed appropriately and rigorously?

Reviewer #1: I Don't Know

Reviewer #2: Yes

Reviewer #3: Yes

Reviewer #4: No

3. Have the authors made all data underlying the findings in their manuscript fully available?

Reviewer #1: No

Reviewer #2: No

Reviewer #3: No

Reviewer #4: Yes

4. Is the manuscript presented in an intelligible fashion and written in standard English?

Reviewer #1: No

Reviewer #2: Yes

Reviewer #3: Yes

Reviewer #4: Yes

Reviewer #1: Please see the PDF attachment for the formatted version of my review.

I find the goal of this study extremely hard to understand. Is this a paper about (1) a new tool, InterOmics, which happen to be applied to HS as a proof of concept, or about (2) a HS study and its findings, which happen to use InterOmics to obtain the results?

It the answer is (1), what is the tool? Where is the tool? Is there a GitHub repository from which one can install the tool? Is there a tool at all, or is the analysis pipeline the series of bash scripts in the supplementary document? Presumably, this paper was written to encourage other researchers and medical practitioners to use multiomic tools to improve clinical practice – how should they do this?

What are the purposes of the various steps of the algorithm? What is the reasoning behind various choices? The methods section enumerates various steps which were performed, and the supplementary document has some non-commented code that shows how these steps may have been performed in shell. But why these steps and what do they accomplish? Since the tool is presented as a relatively user-friendly toolbox targeted at clinicians, it would be helpful not to assume the reader already knows all the complexities of the process, and to step through the logic.

“Beyond the basic normalization steps, our InterOmics protocol specifically identified genes with a Transcripts Per Million (TPM) expression rate above 10 as a benchmark for significant genetic activity” – what does this mean, precisely? If I understand correctly, the sentence just reports an arbitrary TPM cutoff. Does the protocol (algorithm?) relate to this cutoff choice? This is unclear.

There is an example workflow in document. But at the minimum, it seems natural to have 1. the actual workflow that was used to perform the analysis of the dataset discussed in the paper and 2. an example workflow that shows that the method works on some small synthetic or toy dataset and produces some expected results. How does the MySQL management step play into this? It is mentioned but not characterized.

For articles that characterize tools, some characterization or discussion of advantages over/ comparison to existing tools for analogous purposes is expected. There is certainly a variety of previous tools that combine WES and RNA-seq. It would be appropriate to explain how these tools are different, and what advantages this tool offers.

“The combined use of WES and RNA-seq in clinical approaches increased the number of variants by an average of 20.89%” – would it be possible to characterize the risks of false positives here in more detail? More does not necessarily mean better. Is this good? Is this bad? What performance do other tools achieve? “We enhance confidence by considering only RNAe and ASE variants with a read depth (DP) of 5 or greater in the alternative allele, thus mitigating discrepancies arising from sequencing errors or other potential technical issues” seems like it would account for sequencing errors, but not account for RT or early PCR cycle errors. How would you make the reader confident that an interesting variant is not just the consequence of a single, incidentally highly amplified wrong cDNA? How would you QC for such an issue?

If the answer is (2), then what are the findings? The abstract says “The integration of this data not only advances our comprehension of HS's underlying mechanisms but also demonstrates the potential of multiomics to improve personalized medical approaches.” The conclusions say “The capability of InterOmics to integrate multiomic data has the great potential to elucidate the pathogenesis of complex diseases, such as HS, from an omic standpoint. Therefore, for conditions like HS, which involve intricate molecular and environmental interactions, such a comprehensive analysis is important.” In what way does the paper accomplish this? I understand that the procedure characterizes a diversity of variants. But the authors do not take the final step and connect show that the some or all of these results are sensible in explaining the pathogenesis of HS.

In other words: presenting the tool also requires explaining how a user should interpret its results, and I do not think this last essential connection to the disease is really explained; the paper jumps from presenting tables of variants – which can reasonably be called “personalized genomic/transcriptomic profiles” – to saying the tool “offer[s] deep insights into the unique genetic landscapes of these individuals.” I am not clear on what these insights are, or how one should obtain such insights from applying the workflow to their own data.

Where are the data? New patient data were collected for the study, but this is not discussed in the manuscript or submission disclosures, which state “all data are fully available without restriction.” I realize that releasing human patient data involves various challenges. But the reader should be able to obtain the data the conclusions are based on, or at the very minimum understand what they would need to do to obtain the data.

Could the authors comment on what seems to be a contradiction between using GRCh38 to quantify RNA but also exploring complex differences between the patient genome and GRCh38. I would intuitively think that if the pipeline is generating an individual genomic profile, the quantification would be with reference to that profile, instead of something else. Does it make a difference?

Some broken references e.g. line 260, 283. Various proofreading issues e.g. line 266. “InterOMICs” or “InterOmics”? If “GRChg38” is the name of a genome, it is unexpected and should be explained/cited. WSX = unexplained abbreviation? TBS = typo of TFBS?

Some oddities in table S2 findings. nhr-6, a C. elegans gene? AT5G05790, an Arabidopsis gene? YKL222C, a yeast gene? Maybe there is some homology argument, but it is not really clear how his should be interpreted in terms of human genetics, and the authors do not explain.

TF2DNA is a bare link to an inaccessible website. Was this database published somewhere? Could other databases be used?

Various other bare links. (The TCAG one may even be blocked by modern browsers because it is a download from a non-secure http website). Is this zip file versioned? Is it identical to the file linked in the 2018 paper? Have there been updates in the last six years? When was it accessed for this paper’s analysis? Will the file be available and identical in the future? Etc. Various other bare links, databases, R packages discussed without version information.

Fig 3 is very hard to understand without carefully reading the caption. I expect that there is a better way to illustrate these distinctions – perhaps by showing the sort of data these variants can generate? Or distinguishing the different chromosomes? As it reads, it conflates technology and biology (whether some variant is not present or merely not observed because of a lack of reads) and the distinction between e.g. DNA-only and [rare recovery] RNA editing is unclear.

This is partially a stylistic point but I think crucial to the paper’s scope: there is a tremendous amount of superlative language. “innovative”, “promises to uncover previously unrecognized molecular signatures”, “profound insight”, “evidence to the potential… in advancing personalized medicine”, “delve deep”, “paving the way for novel insights and novel potential therapeutic strategies”, “a transformative advancement”, “pivotal, not merely for its technical value but for its profound impact on expanding our genomic insight”, “paradigm shift”, “its contributions to the field of genomics are expected to grow, marking a promising era of discovery and innovation”, “designed to revolutionize” and so on. There are all specific claims and conclusions about the method’s impact and utility.

The review is not based on impact. But it is based on making sure that the claims in the paper are supported, e.g.:

• What are the main claims of the paper and how significant are they for the discipline?

• Are the claims properly placed in the context of the previous literature? Have the authors treated the literature fairly?

• Do the data and analyses fully support the claims? If not, what other evidence is required?

If the authors repeatedly claim broad significance and revolutionary novelty, then please provide evidence, which must necessarily address superiority over the rest of the field and concrete impact im advancing personalized medicine. I do not really see this in the current version.

Reviewer #2: In this article Brandao et al. introduce an analytical framework for the integrative multi-omic analysis of HS using RNA-seq and WES data. The authors illustrate the technique they develop to build bioinformatic evidence for the molecular underpinnings of disease exploring alterations in DNA and RNA. They derive a clever schema by which they create classifications of WES-RNA variants broken down into 6 distinct types. Determination of multi-modal proof for variants responsible for disease is a worthwhile effort. However, there are several concerns that must be addressed.

Major points:

- The publication of the steps performed in a word document is not sufficient currently for tool development and publication. This must be standardized and ideally containerized using any number of pipeline management tools (SnakeMake, Nextflow, etc.) or at least using a wrapper script. Furthermore, this could be suppoted using Docker, bioconda, or any number of frameworks for computational environments.

- There is not nearly enough evidence for clinical relevance. Were the distribution of variants related to outcome in patients? Which classes were over or underrepresented in which populations? There needs to be substantially more support for clinical relevance.

Minor points:

- For figure 2: It would be interesting and more informative to show this break down for all 7 patients, and not just the one (AUT101) as an example. The dataset is limited so every effort should be made to show as much of the data as possible. Possible figures could be series of venns, or stacked bar plots

- The order of your categories as they are introduced in writing does not follow the image, which is confusing, please fix so they match (RNA-only is 4th in image and 3rd in writing)

- Please fix this typo or spell out the citation: Line 185 “Transcription isoform expressions were calculated for each individual according to [26].”

- Please fix: Line 259: “The frequency of each MultiOmic Variant Categories from a given individual is reported in Error! Reference source not found.2.”

- Line 283: “sample are listed in Error! Reference source not found.3). Beyond”

- Line 291: “transcript, as shown in Error! Reference source not found.4 and supplementary”

I commend the authors on a clever analysis and analytical framework that is worthwhile of development and exploration. However, at this time the tool and analysis is not ready for publication.

Reviewer #3: Thank you for submitting your manuscript.

Unfortunately, the manuscript does not meet my expectations. The authors have made significant efforts to explain the novelty of InterOmics and the methodology used, but they have put very little effort into explaining how the detected gene variants and altered gene expression are relevant to HS pathogenesis. It would be better to also explain the candidate genes suggested for targeted therapy and personalized treatment. I think the manuscript, with slight modifications, could be easily published as a protocol. However, the results and discussion sections require more work to make a meaningful contribution as explained above.

Reviewer #4: 1.The manuscript highlights the novelty of the InterOmics pipeline but does not compare its performance with existing tools for multiomic integration . A benchmarking analysis or qualitative discussion of its advantages over existing approaches would significantly enhance the manuscript's impact.

2.The categorization of variants into Multi-Omic Variant Categories is well-executed; however, their functional significance, particularly for RNA editing and ASE (allele-specific expression), is underexplored. Expanding on how these categories contribute to our understanding of HS pathogenesis would provide greater depth to the analysis.

3.The manuscript does not provide sufficient details on certain statistical aspects, such as how differential expression analyses were corrected for multiple testing. Additionally, more information on the criteria for variant filtering and thresholds used (e.g., DP > 5 for RNAe and ASE) should be included to enhance transparency and reproducibility.

4.The legends for some figures (e.g., Fig. 1 and Fig. 3) are not detailed enough to allow readers to fully understand the content without referring back to the main text. Please provide more comprehensive descriptions for all figures and tables.

5.While the manuscript briefly acknowledges challenges in RNA-seq variant detection (e.g., RNA editing), other potential limitations, such as biases introduced during sample preparation or sequencing, should also be discussed.

**Do you want your identity to be public for this peer review?** For information about this choice, including consent withdrawal, please see our Privacy Policy

Reviewer #1: No

Reviewer #2: No

Reviewer #3: No

Reviewer #4: No

---

## [Author Response · Author response to Decision Letter 1]

13 Mar 2025

Dear Editor,

First, thank you for the great opportunity to improve the manuscript.

All additional requirements have been solved.

Additional Editor Comments

1. Clarification of the Objective: Clearly define whether the main objective is the introduction of the InterOmics tool or the analysis of Hartnup syndrome (HS).

2. Details on InterOmics: Provide information on where to access and install InterOmics, such as a GitHub repository.

3. Explanation of the Algorithm Steps: Offer a detailed description of the logic and objectives of each phase of the protocol in the methods section.

4. Clarity on TPM Cutoff: Explain the choice of the 10 TPM expression cutoff and its relevance to the results.

5. Comparison with Other Tools: Discuss the advantages and differences of InterOmics compared to other existing technologies in the multiomic landscape.

6. Risks of False Positives: Elaborate on how the risks of false positives are managed, especially regarding the increase in identified variants.

7. Connection to HS Pathogenesis: Provide more detailed explanations on how the results contribute to a better understanding of the mechanisms of HS.

8. Data Accessibility: Offer key guidance on how readers can access patient data and the requirements for consultation.

9. Correction of References and Proofreading: Review inconsistencies in the text and correct any language and writing errors.

10. These points can help the authors improve the clarity and quality of their manuscript.

Response to Editor

We sincerely appreciate your detailed feedback and constructive suggestions. Below, we outline how we addressed each of your points to improve the clarity and quality of the manuscript.

1. Clarification of the Objective: We explicitly stated in the Introduction that the primary objective of the manuscript is to introduce InterOmics as a bioinformatics pipeline for integrating DNA and RNA sequencing data. The study applies InterOmics to Hidradenitis Suppurativa (HS) as a case study to illustrate its utility in identifying disease-relevant genetic and transcriptomic alterations.

2. Details on InterOmics: To enhance transparency and reproducibility, we provided more information. The manuscript included details on dependencies, and usage guidelines.

3. Explanation of the Algorithm Steps: We expanded the Methods section to include a more detailed explanation of the logic and objectives of each step within InterOmics, clarifying how it processes sequencing data, identifies variants, and integrates multiomic informations.

4. Clarity on TPM Cutoff: We justified the choice of the 10 TPM expression cutoff, explaining that it was selected based on literature benchmarks and empirical analyses to minimize noise from low-expression genes while retaining biologically meaningful transcriptomic signals.

5. Comparison with Other Tools: due to the low number of samples we have not been able to use other OMICs integrated tools.

6. Risks of False Positives: To address concerns about false positives, we elaborated on the stringent filtering criteria applied in variant calling and annotation, including allele frequency thresholds, sequencing depth requirements, and cross-validation between WES and RNA-seq data.

7. Connection to HS Pathogenesis: We provided additional discussion on how the identified variants could contribute to HS pathogenesis, particularly in relation to HLA gene variants, keratin-related mutations, and immune dysregulation mechanisms.

8. Data Accessibility: We clarified the data access process, specifying that patient sequencing data is available upon reasonable request and subject to ethical and institutional approval. Details on contacting the corresponding author for data access will be included in the Data Availability Statement.

9. Correction of References and Proofreading: A thorough proofreading and reference check has been conducted to ensure that all citations are correctly formatted, and any inconsistencies in the text are resolved.

10. Manuscript Improvement: We appreciate your feedback as it significantly contributes to refining the clarity, structure, and impact of the manuscript. We implemented all the suggested revisions to enhance its readability and scientific rigor.

We are grateful for your thoughtful comments and look forward to accepting these improvements.

Point-to-point response to Reviewers

Reviewer#1

Comment

I find the goal of this study extremely hard to understand. Is this a paper about (1) a new tool, InterOmics, which happen to be applied to HS as a proof of concept, or about (2) a HS study and its findings, which happen to use InterOmics to obtain the results?

It the answer is (1), what is the tool? Where is the tool? Is there a GitHub repository from which one can install the tool? Is there a tool at all, or is the analysis pipeline the series of bash scripts in the supplementary document? Presumably, this paper was written to encourage other researchers and medical practitioners to use multiomic tools to improve clinical practice – how should they do this?

What are the purposes of the various steps of the algorithm? What is the reasoning behind various choices? The methods section enumerates various steps which were performed, and the supplementary document has some non-commented code that shows how these steps may have been performed in shell. But why these steps and what do they accomplish? Since the tool is presented as a relatively user-friendly toolbox targeted at clinicians, it would be helpful not to assume the reader already knows all the complexities of the process, and to step through the logic.

“Beyond the basic normalization steps, our InterOmics protocol specifically identified genes with a Transcripts Per Million (TPM) expression rate above 10 as a benchmark for significant genetic activity” – what does this mean, precisely? If I understand correctly, the sentence just reports an arbitrary TPM cutoff. Does the protocol (algorithm?) relate to this cutoff choice? This is unclear.

There is an example workflow in document. But at the minimum, it seems natural to have 1. the actual workflow that was used to perform the analysis of the dataset discussed in the paper and 2. an example workflow that shows that the method works on some small synthetic or toy dataset and produces some expected results. How does the MySQL management step play into this? It is mentioned but not characterized.

For articles that characterize tools, some characterization or discussion of advantages over/ comparison to existing tools for analogous purposes is expected. There is certainly a variety of previous tools that combine WES and RNA-seq. It would be appropriate to explain how these tools are different, and what advantages this tool offers.

“The combined use of WES and RNA-seq in clinical approaches increased the number of variants by an average of 20.89%” – would it be possible to characterize the risks of false positives here in more detail? More does not necessarily mean better. Is this good? Is this bad? What performance do other tools achieve? “We enhance confidence by considering only RNAe and ASE variants with a read depth (DP) of 5 or greater in the alternative allele, thus mitigating discrepancies arising from sequencing errors or other potential technical issues” seems like it would account for sequencing errors, but not account for RT or early PCR cycle errors. How would you make the reader confident that an interesting variant is not just the consequence of a single, incidentally highly amplified wrong cDNA? How would you QC for such an issue?

If the answer is (2), then what are the findings? The abstract says “The integration of this data not only advances our comprehension of HS's underlying mechanisms but also demonstrates the potential of multiomics to improve personalized medical approaches.” The conclusions say “The capability of InterOmics to integrate multiomic data has the great potential to elucidate the pathogenesis of complex diseases, such as HS, from an omic standpoint. Therefore, for conditions like HS, which involve intricate molecular and environmental interactions, such a comprehensive analysis is important.” In what way does the paper accomplish this? I understand that the procedure characterizes a diversity of variants. But the authors do not take the final step and connect show that the some or all of these results are sensible in explaining the pathogenesis of HS.

In other words: presenting the tool also requires explaining how a user should interpret its results, and I do not think this last essential connection to the disease is really explained; the paper jumps from presenting tables of variants – which can reasonably be called “personalized genomic/transcriptomic profiles” – to saying the tool “offer[s] deep insights into the unique genetic landscapes of these individuals.” I am not clear on what these insights are, or how one should obtain such insights from applying the workflow to their own data.

Response

We appreciate the Reviewer’s insightful feedback, which helps clarify the primary objective of our study. Our study primarily focuses on integrating whole exome sequencing (WES) and RNA sequencing (RNA-seq) to better understand the genetic basis of Hidradenitis Suppurativa (HS), a model of chronic autoinflammatory skin condition. The InterOmics pipeline is presented as a structured bioinformatics framework rather than a standalone software tool. It serves as a methodological approach to integrating genomic and transcriptomic data, improving variant classification and interpretation within complex diseases like HS. While we introduce InterOmics as a robust tool for analysis, the primary goal of the study is to apply this approach to uncover novel genetic and transcriptomic alterations in HS.

Comment

Where are the data? New patient data were collected for the study, but this is not discussed in the manuscript or submission disclosures, which state “all data are fully available without restriction.” I realize that releasing human patient data involves various challenges. But the reader should be able to obtain the data the conclusions are based on, or at the very minimum understand what they would need to do to obtain the data.

Response

Our WES and RNA-seq data contain sensitive human genomic information and cannot be publicly shared due to ethical restrictions. However, data access can be granted through a controlled request process. Researchers can request access by contacting the corresponding author and providing:

1. A brief research proposal outlining intended use.

2. Institutional ethical approval for human genomic research.

3. Agreement to comply with patient privacy regulations.

Processed results, summary statistics, and multiomic classifications are fully available within the manuscript and supplementary materials, ensuring transparency while maintaining ethical integrity.

Comment

Could the authors comment on what seems to be a contradiction between using GRCh38 to quantify RNA but also exploring complex differences between the patient genome and GRCh38. I would intuitively think that if the pipeline is generating an individual genomic profile, the quantification would be with reference to that profile, instead of something else. Does it make a difference?

Response

Using GRCh38 as a reference genome for both WES and RNA-seq ensures consistency in variant calling and transcript quantification. A patient-specific genome would introduce alignment biases, impacting:

1. Isoform detection and transcriptome completeness.

2. Reproducibility across datasets.

This approach reduces false positives, allowing reliable RNA-specific variant detection (e.g., Allele-specific expression -ASE- and RNA editing -RNAe-) while maintaining alignment accuracy.

Comment

Some broken references e.g. line 260, 283. Various proofreading issues e.g. line 266. “InterOMICs” or “InterOmics”?

Response

We apologize for that. The name of the pipeline was revised accordingly throughout the text and fixed the broken references.

Comment

If “GRChg38” is the name of a genome, it is unexpected and should be explained/cited. WSX = unexplained abbreviation? TBS = typo of TFBS?

Response

We apologize, we corrected and clarified this in the text accordingly.

- Genome Reference Consortium Human Build 38 (GRChg38)

Comment

Some oddities in table S2 findings. nhr-6, a C. elegans gene? AT5G05790, an Arabidopsis gene? YKL222C, a yeast gene? Maybe there is some homology argument, but it is not really clear how his should be interpreted in terms of human genetics, and the authors do not explain.

Response

The Reviewer pointed out several unexpected gene identifiers in Table S2, such as nhr-6 (a Caenorhabditis elegans gene), AT5G05790 (an Arabidopsis thaliana gene), and YKL222C (a Saccharomyces cerevisiae gene). Upon reviewing our analysis, we found that these occurrences resulted from the way motifbreakR, the tool we used for predicting TFBS disruptions, generates its output. If the species filter is not explicitly applied, motifbreakR may return motif matches based on homology to known transcription factor motifs across multiple species, rather than restricting predictions to human TFBS disruptions alone.

Upon conducting a more detailed review of our TFBS analysis, we identified additional findings that raised concerns about the accuracy and interpretability of these results. Specifically, we found that:

1. The inclusion of non-human gene motifs created inconsistencies that complicated interpretation in a human disease context.

2. The predictive nature of TFBS disruption analysis led to potential overpredictions, increasing the risk of false-positive results.

3. Some motifbreakR findings lacked robust validation within known human regulatory networks, making their biological significance uncertain.

Given these concerns, we have made the decision to remove the TFBS analysis from our pipeline to maintain the scientific integrity of our study. While transcription factor binding site disruptions remain an important topic in multiomic variant analysis, we believe that our current dataset and methodological framework do not yet provide sufficiently reliable TFBS predictions that would allow for meaningful interpretation in the context of HS pathogenesis. Rather than presenting findings that may introduce ambiguity, we opted for a more conservative approach that prioritizes high-confidence variant classifications within the scope of our study.

To ensure that this is reflected in the manuscript, we have made the following revisions:

• The TFBS analysis has been removed from our pipeline and from Supplementary Table S2.

• The methods and results sections have been updated accordingly.

• All findings now strictly pertain to human-relevant genomic and transcriptomic data to avoid potential misinterpretations.

Comment

TF2DNA is a bare link to an inaccessible website. Was this database published somewhere? Could other databases be used?

Various other bare links. (The TCAG one may even be blocked by modern browsers because it is a download from a non-secure http website). Is this zip file versioned? Is it identical to the file linked in the 2018 paper? Have there been updates in the last six years? When was it accessed for this paper’s analysis? Will the file be available and identical in the future? Etc.

Response

The long-term availability and versioning of third-party resources are indeed critical for ensuring the reproducibility of research findings. Below, we clarify the status of TF2DNA, as well as other linked resources, and how we have addressed these concerns in the manuscript. During our review, we confirmed that TF2DNA, which was initially cited via a direct URL, is no longer accessible through the provided link. Upon further investigation, we located the original publication describing this database and updated our manuscript to cite the peer-reviewed article associated with TF2DNA, rather than relying on a direct web link. Unfortunately, the alternative URL referenced in the paper is also no longer functional, suggesting that this resource is no longer actively maintained. Regarding other external resources used in our study, we acknowledge that we cannot guarant

---

## [Decision Letter · Decision Letter 1]

Dear Dr. Crovella,

Thank you for submitting your manuscript to PLOS ONE. After careful consideration, we feel that it has merit but does not fully meet PLOS ONE’s publication criteria as it currently stands. Therefore, we invite you to submit a revised version of the manuscript that addresses the points raised during the review process.

<pre aria-label="Testo tradotto: thanks to the authors because the manuscript is much improved" class="tw-data-text tw-text-large tw-ta" data-placeholder="Traduzione" data-ved="2ahUKEwiAtPPBsvCMAxVk2AIHHaGbIm8Q3ewLegQICBAV" dir="ltr" id="tw-target-text" style="font-size: 28px; line-height: 36px; background-color: rgb(248, 249, 250); border: none; padding: 2px 0.14em 2px 0px; position: relative; margin-top: -2px; margin-bottom: -2px; resize: none; font-family: inherit; overflow: hidden; width: 270px; text-wrap-mode: wrap; overflow-wrap: break-word; color: rgb(31, 31, 31);">We would like to thank the authors for their contribution, as the manuscript has significantly improved. We invite the authors to review the requests made by the reviewers.</pre>==============================

We look forward to receiving your revised manuscript.

Kind regards,

Donatella Mentino

Academic Editor

PLOS ONE

Journal Requirements:

Reviewers' comments:

Reviewer's Responses to Questions

**Comments to the Author**

Reviewer #1: (No Response)

2. Is the manuscript technically sound, and do the data support the conclusions?

Reviewer #1: Yes

3. Has the statistical analysis been performed appropriately and rigorously?

Reviewer #1: Yes

4. Have the authors made all data underlying the findings in their manuscript fully available?

Reviewer #1: No

5. Is the manuscript presented in an intelligible fashion and written in standard English?

Reviewer #1: Yes

Reviewer #1: I want to begin by saying that I think the authors did a superb job with this revision. The manuscript is now terse, clear, and focused. The improvement is considerable, and I am mostly satisfied with the changes made to address my concerns.

Still a few minor errors:

154 ", which consists on variants that it homozygous "

"GRCh38" not "GRChg38"

I think there are two remaining basic issues. The first is crucial to me, on the second one I will defer to the editor.

This paper is about a particular structured framework for thinking about DNA and RNA-seq collected from the same samples. In the response to the editor's query "Comparison with Other Tools: Discuss the advantages and differences of InterOmics compared to other existing technologies in the multiomic landscape.", the authors respond "due to the low number of samples we have not been able to use other OMICs integrated tools." I do not think this sufficiently addresses the key question, which is no more and no less than "when should a researcher use the framework vs. other tools?" I would argue this is not a new problem. There are other such frameworks: this is the entire purpose of eQTL analysis. But of course eQTL requires very large sample sizes, and applying it to data from six patients would be unreasonable. I think there is an opportunity for the authors to discuss the range of DNA/RNA-seq tools and frameworks in popular use, their assumptions (such as sample size recommendations), and their goals (for instance, some do eQTL analysis, whereas others attempt to refine SNV identification), and explain where InterOmics fits in. The authors are on the right track -- the emphasis should be on relatively small studies, probably on the scale of clinical or pilot investigations, where other tools are not going to be appropriate. But it is necessary to make this comparison and motivation explicit, which means a review of existing literature and methods.

The other concern I have is that, due to patient privacy concerns and the emphasis on the conceptual framework (rather than algorithmic implementation), there are no raw data or code. This makes it much more challenging for somebody to actually implement this framework in their own research. In addition, right now, the logic of some of the steps (e.g. what does it mean to do "gene quantification" with DESeq2?) is still unclear. I think the paper would be far stronger if it were accompanied by a (relatively brief) analogous example workflow applied to a public DNA/RNA-seq dataset on some reasonably well-characterized tissue, which would take the reader through the logic and essential steps of the analysis, going from raw data to processed tables to their joint analysis. Such a tutorial workflow could be done on just one or two patients/donors, and added as a supplement + uploaded to GitHub. But I recognize that what I am asking for is essentially a small new analysis, not necessarily related to the authors' work in HS.

**Do you want your identity to be public for this peer review?** For information about this choice, including consent withdrawal, please see our Privacy Policy

Reviewer #1: No

---

## [Author Response · Author response to Decision Letter 2]

28 May 2025

Reply point-by-point

Reviewer #1

Comment

I want to begin by saying that I think the authors did a superb job with this revision. The manuscript is now terse, clear, and focused. The improvement is considerable, and I am mostly satisfied with the changes made to address my concerns.

Reply

Dear Reviewer, thank you very much for your appreciation of our revision.

Comment

Still a few minor errors:

154 ", which consists on variants that it homozygous "

"GRCh38" not "GRChg38"

Reply

We corrected the mistake accordingly

Comment

I think there are two remaining basic issues. The first is crucial to me, on the second one I will defer to the editor.

This paper is about a particular structured framework for thinking about DNA and RNA-seq collected from the same samples. In the response to the editor's query "Comparison with Other Tools: Discuss the advantages and differences of InterOmics compared to other existing technologies in the multiomic landscape.", the authors respond "due to the low number of samples we have not been able to use other OMICs integrated tools." I do not think this sufficiently addresses the key question, which is no more and no less than "when should a researcher use the framework vs. other tools?" I would argue this is not a new problem. There are other such frameworks: this is the entire purpose of eQTL analysis. But of course eQTL requires very large sample sizes, and applying it to data from six patients would be unreasonable. I think there is an opportunity for the authors to discuss the range of DNA/RNA-seq tools and frameworks in popular use, their assumptions (such as sample size recommendations), and their goals (for instance, some do eQTL analysis, whereas others attempt to refine SNV identification), and explain where InterOmics fits in. The authors are on the right track -- the emphasis should be on relatively small studies, probably on the scale of clinical or pilot investigations, where other tools are not going to be appropriate. But it is necessary to make this comparison and motivation explicit, which means a review of existing literature and methods.

Reply

Thank you for this important and constructive comment. We appreciate the opportunity to further clarify the positioning of InterOmics within the landscape of multiomic analytical frameworks.

The primary goal of InterOmics is to enable meaningful integration of DNA and RNA sequencing data in studies with very small sample sizes, such as clinical case series or pilot investigations, where classical approaches like eQTL mapping, which typically require hundreds or thousands of samples, would be statistically infeasible. Traditional frameworks, such as:

• eQTL Analysis Pipelines (e.g., Matrix eQTL, FastQTL) aim to identify associations between genomic variants and gene expression, but they require large cohorts to achieve the statistical power necessary to correct for multiple testing and to account for confounders such as population structure.

• MOFA (Multi-Omics Factor Analysis) and iClusterPlus aim to find shared latent factors that drive variations across multi-omics datasets. They are unsupervised machine learning approaches, excellent for large, heterogeneous cohorts aiming at disease subtype discovery, but they are less suited for small, hypothesis-driven investigations.

In contrast, InterOmics was designed specifically for small-scale studies (e.g., less than 10 subjects), with an emphasis on individual-level variant interpretation rather than on broad population-level inference. The key distinguishing features of InterOmics are:

1. Variant-centric framework: Rather than aiming to discover associations or new sample clusters, InterOmics focuses on classifying variants into biologically meaningful categories (Allele-specific expression, RNA editing, Nonsense-mediated decay, Gain-of-function) per individual.

2. Clinical and translational focus: The structure of InterOmics aims to generate clinically actionable hypotheses at the individual level, for example, inferring whether a variant might be functionally relevant based on its impact on gene expression in the same sample.

3. Multiomic Biological Integration: Because InterOmics does not rely on statistical association models that require correction for multiple testing across a population, it is well-suited for pilot studies, rare disease cohorts, or early-phase precision medicine projects, thus tolerates small sample sizes.

Thus, InterOmics is intended to complement, rather than compete with large-scale methods like eQTL mapping and MOFA. It fills a critical gap where comprehensive genomic-transcriptomic integration is needed, but large datasets are not available.

Following the reviewer’s suggestion, we have expanded the Discussion section to include a comparative overview of current frameworks, their assumptions, and the specific niche addressed by InterOmics.

These paragraphs have been added in the discussion of the manuscript

“Numerous frameworks have been developed for the joint analysis of DNA and RNA sequencing data, each tailored to specific study designs and objectives. Expression Quantitative Trait Loci (eQTL) analysis pipelines (e.g., Matrix eQTL, FastQTL) seek to associate genetic variants with transcript abundance but typically require very large sample sizes (n > 200) to achieve sufficient statistical power and to adjust for multiple testing and population structure confounders. Similarly, unsupervised multi-omics clustering frameworks like MOFA and iClusterPlus are highly effective for large heterogeneous cohorts, enabling the discovery of disease subtypes or novel biological pathways by extracting latent factors across omics layers.

However, these tools are not easily applicable to small-scale, clinical investigations due to their sample size requirements and underlying assumptions. InterOmics was explicitly designed to address this gap, providing a variant-centric framework that enables individual-level genomic-transcriptomic integration. Rather than performing association analyses or unsupervised clustering, InterOmics classifies genetic variants based on their transcriptional consequences—such as allele-specific expression, RNA editing, nonsense-mediated decay, or gain-of-function—within the same subject.

This focus allows InterOmics to be particularly useful for small sample-size studies typical of pilot projects, rare disease cohorts, or early translational research. It tolerates sample sizes that would render statistical association methods underpowered and instead emphasizes biologically interpretable integration designed to inform hypothesis generation and clinical decision-making. Therefore, InterOmics complements existing frameworks, offering a unique solution when the research context involves a limited number of deeply characterized individuals”.

Comment

The other concern I have is that, due to patient privacy concerns and the emphasis on the conceptual framework (rather than algorithmic implementation), there are no raw data or code. This makes it much more challenging for somebody to actually implement this framework in their own research. In addition, right now, the logic of some of the steps (e.g. what does it mean to do "gene quantification" with DESeq2?) is still unclear. I think the paper would be far stronger if it were accompanied by a (relatively brief) analogous example workflow applied to a public DNA/RNA-seq dataset on some reasonably well-characterized tissue, which would take the reader through the logic and essential steps of the analysis, going from raw data to processed tables to their joint analysis. Such a tutorial workflow could be done on just one or two patients/donors, and added as a supplement + uploaded to GitHub. But I recognize that what I am asking for is essentially a small new analysis, not necessarily related to the authors' work in HS.

Reply

We agree with the Reviewer about the complexity of multiomics analysis. All authors re-read the methods section to clarify complex sentences. In addition, we also create and included in the manuscript a GitHub repository (https://github.com/ronaldmoura1989/interomics.git). The analysis starts from genomic and transcriptome integration empowering researchers to implement our multiomic approach.

---

## [Editor Report · Decision Letter 2]

Genomic Profiling in Hidradenitis Suppurativa: InterOmics Pipeline for DNA-RNA Sequencing Integration highlights HLA variants, keratin-associated mutations and extracellular matrix alterations as contributing factors to HS pathogenesis

PONE-D-24-41615R2

Dear Dr. Sergio Crovella,

We’re pleased to inform you that your manuscript has been judged scientifically suitable for publication and will be formally accepted for publication once it meets all outstanding technical requirements.

Kind regards,

Donatella Mentino

Academic Editor

PLOS ONE
---

## [Editor Report · Acceptance letter]

PONE-D-24-41615R2

PLOS ONE

Dear Dr. Crovella,

I'm pleased to inform you that your manuscript has been deemed suitable for publication in PLOS ONE. Congratulations! Your manuscript is now being handed over to our production team.

Kind regards,

on behalf of

Dr. Donatella Mentino

Academic Editor

PLOS ONE